# Association of the Hemoglobin to Serum Creatinine Ratio with In-Hospital Adverse Outcomes after Percutaneous Coronary Intervention among Non-Dialysis Patients: Insights from a Japanese Nationwide Registry (J-PCI Registry)

**DOI:** 10.3390/jcm9113612

**Published:** 2020-11-10

**Authors:** Yohei Numasawa, Taku Inohara, Hideki Ishii, Kyohei Yamaji, Shun Kohsaka, Mitsuaki Sawano, Masaki Kodaira, Shiro Uemura, Kazushige Kadota, Tetsuya Amano, Masato Nakamura, Yuji Ikari

**Affiliations:** 1Department of Cardiology, Japanese Red Cross Ashikaga Hospital, 284-1 Yobe-cho, Ashikaga, Tochigi 326-0843, Japan; mskodaira@gmail.com; 2Department of Cardiology, Keio University School of Medicine, Tokyo 160-8582, Japan; taku.inohara@gmail.com (T.I.); sk@keio.jp (S.K.); mitsuakisawano@gmail.com (M.S.); 3Department of Cardiology, Fujita Health University Bantane Hospital, Nagoya 454-8509, Japan; hkishii@med.nagoya-u.ac.jp; 4Department of Cardiology, Kokura Memorial Hospital, Kitakyushu 802-8555, Japan; kyohei@yamaji.info; 5Department of Cardiology, Kawasaki Medical School, Kurashiki 701-0192, Japan; suemura@med.kawasaki-m.ac.jp; 6Department of Cardiology, Kurashiki Central Hospital, Kurashiki 710-8602, Japan; kk5617@kchnet.or.jp; 7Department of Cardiology, Aichi Medical University, Nagakute 480-1195, Japan; amanotaha@yahoo.co.jp; 8Division of Cardiovascular Medicine, Toho University Ohashi Medical Center, Tokyo 153-8515, Japan; masato@oha.toho-u.ac.jp; 9Division of Cardiology, Tokai University School of Medicine, Isehara 259-1193, Japan; ikari@is.icc.u-tokai.ac.jp

**Keywords:** hemoglobin, anemia, creatinine, chronic kidney disease, percutaneous coronary intervention

## Abstract

Although baseline hemoglobin and renal function are both important predictors of adverse outcomes after percutaneous coronary intervention (PCI), scarce data exist regarding the combined impact of these factors on outcomes. We sought to investigate the impact and threshold value of the hemoglobin to creatinine (Hgb/Cr) ratio, on in-hospital adverse outcomes among non-dialysis patients in a Japanese nationwide registry. We analyzed 157,978 non-dialysis patients who underwent PCI in 884 Japanese medical institutions in 2017. We studied differences in baseline characteristics and in-hospital clinical outcomes among four groups according to their quartiles of the Hgb/Cr ratios. Compared with patients with higher Hgb/Cr ratios, patients with lower ratios were older and had more comorbidities and complex coronary artery disease. Patients with lower hemoglobin and higher creatinine levels had a higher rate of in-hospital adverse outcomes including in-hospital mortality and procedural complications (defined as occurrence of cardiac tamponade, cardiogenic shock after PCI, emergency operation, or bleeding complications that required blood transfusion). On multivariate analyses, Hgb/Cr ratio was inversely associated with in-hospital mortality (odds ratio: 0.91, 95% confidence interval: 0.89–0.92; *p* < 0.001) and bleeding complications (odds ratio: 0.92, 95% confidence interval: 0.90–0.94; *p* < 0.001). Spline curve analysis demonstrated that these risks started to increase when the Hgb/Cr ratio was <15, and elevated exponentially when the ratio was <10. Hgb/Cr ratio is a simple index among non-dialysis patients and is inversely associated with in-hospital mortality and bleeding complications after PCI.

## 1. Introduction

Previous studies reported that preprocedural anemia [1,2,3,4,5,6,7] and impaired renal function [8,9,10,11,12,13,14] are both associated with adverse outcomes such as death and bleeding after percutaneous coronary intervention (PCI). Indeed, most of the established risk prediction models for mortality and bleeding include these two factors as dependent variables [15,16]. However, worsening renal function can directly cause anemia (e.g., renal anemia), and low baseline hemoglobin can aggravate renal dysfunction [1,2,3,4,17,18]. Previous observational studies have shown that these two factors may additively or synergistically affect clinical outcomes after PCI [17,18,19,20], although the combined impact of preprocedural hemoglobin and renal function on clinical outcomes has not been thoroughly investigated, especially in non-dialysis patients. Furthermore, relatively few data exist regarding the relationship between the composite indicator of these factors and adverse outcomes after PCI [21].

Hemoglobin and serum creatinine levels are readily available parameters prior to PCI procedures. Accordingly, we sought to develop a simple composite indicator of baseline hemoglobin and serum creatinine levels to investigate their compound effect on clinical outcomes, and combined these two factors into a single index, the hemoglobin to creatinine (Hgb/Cr) ratio, in a contemporary Japanese nationwide coronary intervention registry (J-PCI registry). The primary aim of this study was to assess the impact of the Hgb/Cr ratio on in-hospital clinical outcomes after PCI among non-dialysis patients. As a secondary aim, we sought to clarify the threshold value of the Hgb/Cr ratio, below which the risk of in-hospital adverse outcomes increases, using a spline curve analysis.

## 2. Methods

For the present analysis, we extracted the patient-based data from the ongoing J-PCI registry. The overview of this Japanese nationwide PCI registry has been previously reported in detail [14,22,23,24,25,26,27]. Briefly, the J-PCI registry is a prospective, multicenter registry of the Japanese Association of Cardiovascular Intervention and Therapeutics (CVIT), that aims to collect the characteristics and clinical outcomes of patients undergoing PCI. The protocol of the J-PCI registry was approved by the Institutional Review Board Committee at the Network for Promotion of Clinical Studies in a specified non-profit organization affiliated with Osaka University Graduate School of Medicine (CRPNJSOP-4-5). Additionally, this study complied with the principles of the Declaration of Helsinki.

This study involved patients who underwent PCI and who were recorded in the J-PCI registry from January to December 2017. The J-PCI registry included 260,242 patients who underwent PCI during this period. Patients who were <20 or >100 years of age and those with missing data for preprocedural hemoglobin and/or serum creatinine levels and in-hospital outcomes were excluded. Because the serum creatinine levels in dialysis patients fluctuate before and after hemodialysis, and it is clear that patients with advanced chronic kidney disease (CKD) have worse clinical outcomes after PCI than those without [8,9,10,11,12,13,14], in this study we excluded dialysis patients and patients with baseline serum creatinine levels >3.0 mg/dL. In addition, baseline hemoglobin levels <3.0 g/dL or >20.0 g/dL were also excluded as potential outliers. Consequently, a total of 157,978 non-dialysis patients who underwent PCI for acute coronary syndrome (ACS) and stable coronary artery disease (CAD) from 884 Japanese medical institutions were finally included in this study (Figure 1).

In the J-PCI data dictionary, patients with ACS were defined as those with ST-elevation myocardial infarction (STEMI), non-STEMI, and unstable angina, and patients with stable CAD were defined as those with non-ACS CAD such as stable angina, old myocardial infarction, and silent ischemia. Cardiogenic shock was defined as a sustained condition of systolic blood pressure <80 mmHg, and/or a cardiac index of <1.8 L/min/m^2^ identified to be secondary to cardiac dysfunction, and/or the need for parenteral inotropic or vasopressor agents or mechanical supports. Acute heart failure was defined as heart failure symptoms within 24 h prior to PCI, which corresponded to heart failure of New York Heart Association Functional Classification Class IV. CKD was defined as the presence of proteinuria, or serum creatinine ≥1.3 mg/dL, or an estimated glomerular filtration rate ≤60 mL/min/1.73 m^2^ [14,22]. Successful PCI was defined as achievement of Thrombolysis in Myocardial Infarction (TIMI) flow grade III with residual stenosis ≤25% in the target lesion.

In-hospital complications were defined as in-hospital death within 30 days after PCI, cardiac tamponade, cardiogenic shock during and after PCI, emergency operations, and bleeding complications. A bleeding complication was defined as an access-site or non-access-site bleeding event that required blood transfusion.

Patients were categorized into four groups (quartiles) according to their Hgb/Cr ratio (Figure 1). Regarding the statistical analysis, continuous variables are presented as mean ± standard deviation, and categorical variables are presented as frequency and percentage. For comparisons of patients’ data among groups, we used analysis of variance for continuous variables and Pearson’s chi-square test for categorical variables. Logistic random-effects regression models were used to determine the independent predictors of in-hospital mortality and bleeding complications. For statistical adjustment, we included the following variables: age, Hgb/Cr ratio, oral anticoagulants, female sex, history of heart failure, acute heart failure, STEMI, non-STEMI, cardiogenic shock, hypertension, hyperlipidemia, diabetes mellitus, three-vessel disease, left main trunk (LMT) lesions, and access site (using transfemoral intervention as a reference). In addition, cubic spline curves for the Hgb/Cr ratio were described to identify the inflection points regarding the risks for in-hospital mortality and bleeding complications. We included institutes as a random intercept in all multivariable models. All statistical analyses were performed using R statistical software, version 3.3.3 (R Foundation for Statistical Computing, Vienna, Austria). A *p* value < 0.05 was considered statistically significant.

## 3. Results

The baseline clinical characteristics of the 157,978 study patients who underwent PCI for ACS or stable CAD, stratified by the Hgb/Cr ratio quartiles are shown in Table 1. Patients’ mean age was 70.6 years; 60,594 patients (38.4%) underwent PCI for ACS, and 97,384 patients (61.6%) underwent PCI for stable CAD. The average baseline hemoglobin, serum creatinine, and Hgb/Cr ratio in the total cohort was 13.3 g/dL, 0.9 mg/dL, and 15.9, respectively.

Compared with patients with higher Hgb/Cr ratios, those with lower ratios were older and had more comorbidities, namely, hypertension, diabetes mellitus, a history of PCI, coronary artery bypass grafting, heart failure and myocardial infarction, peripheral artery disease, chronic obstructive pulmonary disease, and CKD. Conversely, patients with higher Hgb/Cr ratios had a higher prevalence of hyperlipidemia and smoking habit. Patients with lower Hgb/Cr ratios had a higher incidence of cardiogenic shock, acute heart failure, and cardiopulmonary arrest on admission than those with higher levels. Regarding the angiographic data, patients with lower Hgb/Cr ratios had more complex lesions such as multi-vessel disease and LMT lesions than those with higher ratios. Notably, patients with lower Hgb/Cr ratios took oral anticoagulants more frequently than those with higher ratios.

The in-hospital clinical outcomes of the patients are shown in Table 2. The procedural success rate was significantly lower in patients with lower Hgb/Cr ratios than in those with higher ratios. Furthermore, in-hospital adverse events were observed more frequently in patients with lower vs. higher Hgb/Cr ratios regarding in-hospital mortality, cardiac tamponade, cardiogenic shock after PCI, emergency operation, and bleeding complications. The rates of in-hospital mortality and bleeding complications according to preprocedural hemoglobin and creatinine quartiles are shown in Figure 2 and Figure 3. Patients with lower hemoglobin and higher creatinine levels tended to have higher rates of in-hospital mortality and bleeding complications. Specifically, patients in the lowest hemoglobin quartile and the highest creatinine quartile had the worst in-hospital outcomes after PCI.

The results of the multivariable logistic regression analyses for in-hospital mortality and bleeding complications are shown in Table 3. Hgb/Cr ratio was inversely associated with these adverse clinical outcomes, when it was entered as a continuous variable. In addition to Hgb/Cr ratio, age, female sex, history of heart failure, acute heart failure, acute myocardial infarction, cardiogenic shock, hypertension, hyperlipidemia, three-vessel disease, and LMT lesions were independent predictors of in-hospital mortality, whereas age, oral anticoagulants, female sex, acute myocardial infarction, cardiogenic shock, hyperlipidemia, diabetes mellitus, and LMT lesions were independently associated with bleeding complications. Importantly, transradial intervention was inversely associated with both in-hospital mortality and bleeding complications. Spline curves demonstrated a non-linear relationship between Hgb/Cr ratio and in-hospital adverse outcomes (Figure 4). The risks of in-hospital mortality and bleeding complications started to increase when the Hgb/Cr ratio was <15, and increased exponentially when the ratio was <10.

## 4. Discussion

The present study demonstrated the impact of Hgb/Cr ratio on in-hospital adverse outcomes after PCI among non-dialysis patients in a Japanese large prospective nationwide registry. The Hgb/Cr ratio, which is user-friendly at bedside because of its simplicity, was inversely associated with in-hospital mortality and bleeding complications. Non-dialysis patients with lower Hgb/Cr ratios, especially those with Hgb/Cr ratios <10, carried a greater risk of in-hospital death and bleeding after PCI.

Prior studies showed that lower preprocedural hemoglobin and impaired renal function are associated with worse clinical outcomes after PCI [1,2,3,4,5,6,7,8,9,10,11,12,13,14]. These two factors are crucial for precise risk assessment for adverse outcomes after PCI, namely, mortality and bleeding [15,16], but the combined effects of anemia and impaired renal function have not been thoroughly investigated, especially in contemporary and large PCI registries. Generally, patients with CKD tend to have lower hemoglobin levels compared with those without, secondary to renal anemia, even in patients with CAD who undergo PCI [1,2,3,4,17,18]. However, some anemic patients have normal renal function, and some patients with CKD have normal hemoglobin concentration, as shown in this study. Observational studies have reported that the presence of anemia and impaired renal function cumulatively affect adverse outcomes after PCI [17,18,19,20]. Therefore, the composite indicator of baseline hemoglobin and renal function may be more important and useful for risk stratification than individual parameters.

Giraldez et al. [21] reported that the laboratory index (15 − hemoglobin [g/dL] + (100 − creatinine clearance [mL/min])/8) was a powerful tool to predict death in patients with STEMI. Although this laboratory index, which includes both baseline hemoglobin and renal function, had sufficient discriminatory capacity, the formula is slightly complex at bedside. In addition, the originating study was performed during the era of thrombolytic therapy, and only 19.5% of the patients underwent PCI during hospitalization. Moreover, because information regarding a patient’s body weight is sometimes lacking, especially in patients presenting with STEMI, creatinine clearance cannot be calculated accurately using the Cockcroft–Gault equation [28] in these patients. Creatinine clearance [8,9,21] or estimated glomerular filtration rate [10,11,12,13] indicate patients’ renal function more precisely than serum creatinine level, but these laboratory values are not always user-friendly because of the complexity of the calculations. Previous studies have also reported detailed risk prediction models for mortality and bleeding after PCI that include factors related to preprocedural hemoglobin and renal function [15,16]. These established risk models are critically important, but for interventional cardiologists, these models are also too complex and time-consuming at bedside, especially in advance of urgent PCI procedures. Therefore, using these risk prediction models is not common in clinical practice. A risk stratification tool must be simple and easily applicable, even in emergency situations. From this perspective, the Hgb/Cr ratio may be a useful tool at bedside because of its simplicity and user-friendliness because hemoglobin and serum creatinine levels are routinely obtained parameters, and the ratio can be roughly calculated mentally, without calculators.

The reasons for worse clinical outcomes after PCI in patients with lower hemoglobin and higher creatinine levels are thought to be multifactorial [3,14]. First, the presence of anemia and CKD may be directly associated with the progression of CAD and myocardial ischemia [3,10,14]. Second, as shown in this study, patients with anemia and CKD tend to be older and have more comorbidities such as hypertension, diabetes mellitus, heart failure, peripheral artery disease, and chronic obstructive pulmonary disease than those without [1,2,3,4,6,7,8,9,10,11,12,13,14,18,19,20,21]. Furthermore, patients with lower Hgb/Cr ratios had a higher incidence of cardiogenic shock, acute heart failure, and cardiopulmonary arrest on admission than those with higher ratios in this study, and this trend was consistent with previous studies [3,8,9,14,21]. Whether the presence of anemia and CKD is a reflection of such comorbidities and/or serious conditions in older patients remains uncertain, but these factors are at least important surrogate markers of adverse outcomes after PCI. Third, patients with anemia and CKD may have a hypercoagulable condition, which can increase the thrombotic risk [2,4,10]. Moreover, activation of sympathetic nerves and the renin-angiotensin-aldosterone system, chronic inflammation, oxidative stress, and endothelial dysfunction may also be associated with adverse clinical outcomes in patients with anemia and CKD [3,13,14,17].

In our multivariable analyses, the Hgb/Cr ratio was inversely associated with in-hospital adverse outcomes. Furthermore, the relationships between the Hgb/Cr ratio and in-hospital adverse outcomes were not linear, but curvilinear. Regarding the Hgb/Cr threshold value, spline curves demonstrated that the risks of in-hospital mortality and bleeding complications increased slightly with Hgb/Cr ratios <15 and increased dramatically with Hgb/Cr ratios <10. These results indicated that patients with Hgb/Cr ratios <10 were the most vulnerable subset and require utmost caution during and after PCI, and those with Hgb/Cr ratios from 10 to 15 were considered to have borderline risks. Overall, the results of this study demonstrated that anemia and CKD were cumulatively associated with in-hospital adverse outcomes after PCI. Accordingly, physicians should pay attention to even mild anemia and mild CKD because the coexistence of these factors may additionally worsen patients’ outcomes [20,21].

Risk assessment is crucial prior to coronary intervention. Our data provide additional insights regarding preprocedural risk stratification using the simple composite indicator of hemoglobin and serum creatinine among non-dialysis patients who undergo PCI. Although effective treatment strategies for underlying anemia and CKD prior to PCI procedures remain unknown [3,5,20], an initial risk assessment according to the preprocedural hemoglobin and serum creatinine levels should be performed in patients undergoing PCI irrespective of their clinical presentation with ACS or stable CAD.

The present study had several limitations. First, although we performed multivariable logistic regression analyses to adjust for covariate imbalances, residual unmeasured confounding factors such as frailty, cognitive function, and socioeconomic status might have affected the outcomes. Second, we had no precise data regarding the individual etiology of anemia and CKD in patients with low hemoglobin and high creatinine levels. Third, lack of long-term follow-up data regarding not only clinical outcomes, but also hemoglobin and creatinine levels after PCI were also important limitations. Serial changes in these parameters are important and would affect the outcome or prognosis in patients undergoing PCI, especially those with acute coronary syndrome and/or acute kidney injury [29,30]. Fourth, precise information regarding medical therapies except antithrombotic agents were lacking in this registry. Finally, the definition of bleeding complications was different from standardized definitions such as the Bleeding Academic Research Consortium criteria [31], which may have affected the study results.

## 5. Conclusions

The Hgb/Cr ratio is a simple index for use among non-dialysis patients that is inversely associated with in-hospital mortality and bleeding complications after PCI. Patients with lower Hgb/Cr ratios, especially those with Hgb/Cr ratios <10, carry a greater risk of in-hospital death and bleeding after PCI.

## Figures and Tables

**Figure 1 jcm-09-03612-f001:**
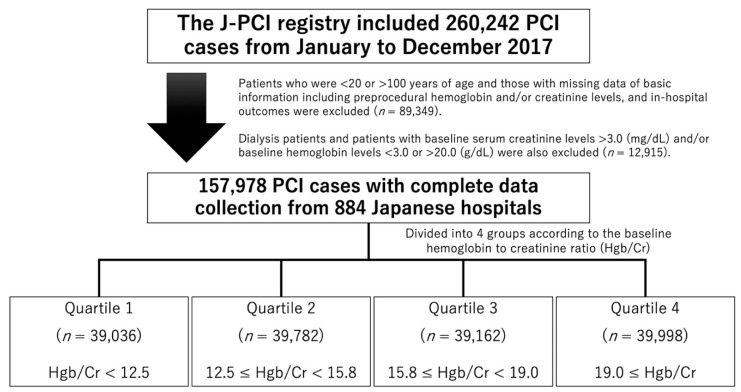
Flow chart of study enrollment. J-PCI registry = Japanese nationwide percutaneous coronary intervention registry, Hgb = hemoglobin, Cr = creatinine.

**Figure 2 jcm-09-03612-f002:**
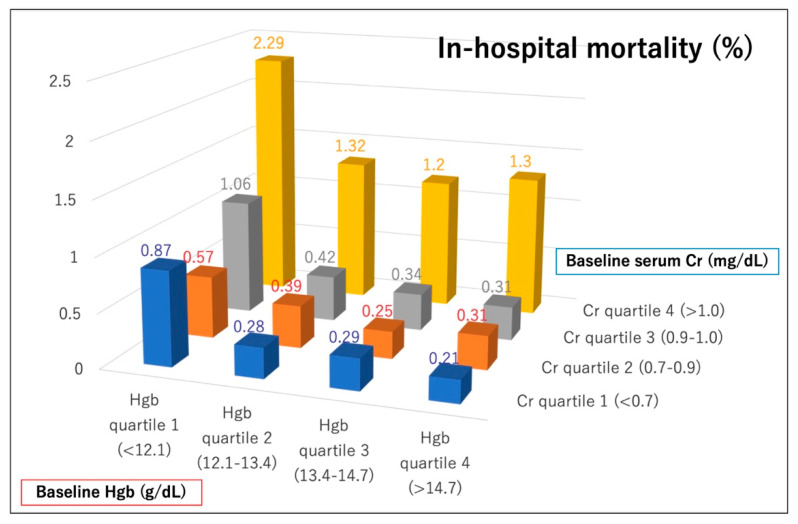
In-hospital mortality according to baseline hemoglobin and creatinine quartiles. Hgb = hemoglobin, Cr = creatinine.

**Figure 3 jcm-09-03612-f003:**
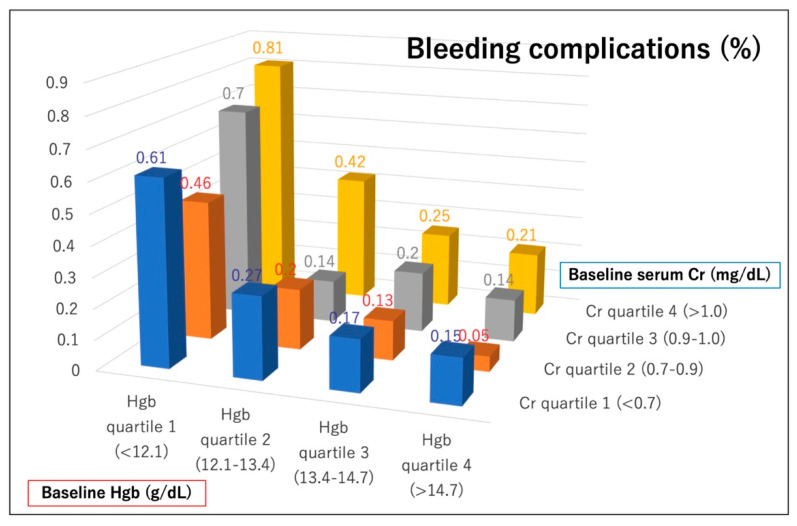
Incidence of bleeding complications according to baseline hemoglobin and creatinine quartiles. Hgb = hemoglobin, Cr = creatinine.

**Figure 4 jcm-09-03612-f004:**
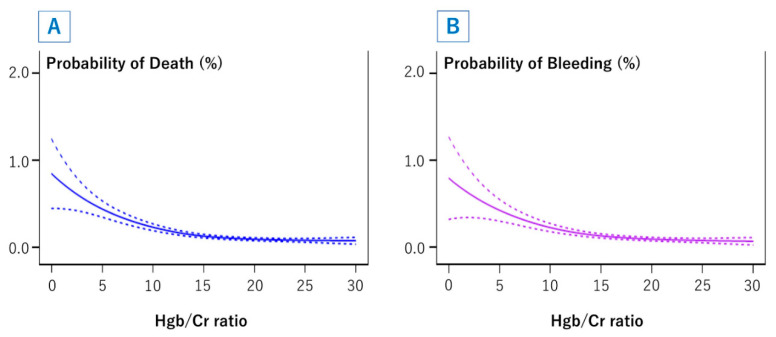
Cubic spline models for in-hospital mortality (**A**) and bleeding complications (**B**) according to the hemoglobin to creatinine ratio. Hgb = hemoglobin, Cr = creatinine. Separate cubic spline modeling was performed for in-hospital mortality (**A**) and bleeding complications (**B**) according to the hemoglobin to creatinine ratio. The smoothed spline plots are shown as solid lines, and upper and lower 95% confidence intervals are shown as dotted lines. Adjusted models included age, hemoglobin to creatinine ratio, oral anticoagulants, female sex, history of heart failure, acute heart failure, ST-elevation myocardial infarction, non-ST-elevation myocardial infarction, cardiogenic shock, diabetes mellitus, three-vessel disease, left main trunk lesion, and access site.

**Table 1 jcm-09-03612-t001:** Baseline clinical characteristics.

Characteristics		Hemoglobin to Creatinine Ratio (Hemoglobin [g/dL]/Creatinine [mg/dL])				
	Total Cohort	Quartile 1 <12.5	Quartile 2 12.5 ≤ … < 15.8	Quartile 3 15.8 ≤ … <19.0	Quartile 4 19.0≤	*p*-Value
	*n* = 157,978	*n* = 39,036	*n* = 39,782	*n* = 39,162	*n* = 39,998	
Age, (years)	70.6 ± 11.2	75.7 ± 9.7	71.5 ± 10.4	68.6 ± 11.1	66.7 ± 11.6	<0.001
Sex, female	36,768 (23.3)	8098 (20.7)	7014 (17.6)	7888 (20.1)	13,768 (34.4)	<0.001
Hypertension	117,034 (77.2)	31,296 (82.7)	29,830 (78.3)	28,062 (75.0)	27,846 (73.0)	<0.001
Hyperlipidemia	103,908 (68.6)	23,906 (63.1)	26,188 (68.7)	26,655 (71.2)	27,159 (71.2)	<0.001
Diabetes mellitus	67,231 (44.4)	18,942 (50.0)	16,488 (43.3)	15,416 (41.2)	16,385 (43.0)	<0.001
Current smoker	50,159 (33.1)	10,525 (27.8)	12,375 (32.5)	13,165 (35.2)	14,094 (37.0)	<0.001
History of percutaneous coronary intervention	72,780 (46.2)	19,888 (51.1)	19,971 (50.3)	17,932 (45.9)	14,989 (37.5)	<0.001
History of coronary artery bypass grafting	5056 (3.2)	2015 (5.2)	1358 (3.4)	999 (2.6)	684 (1.7)	<0.001
History of heart failure	20,476 (13.1)	9563 (24.8)	4937 (12.5)	3359 (8.7)	2617 (6.6)	<0.001
History of myocardial infarction	37,144 (23.7)	10,947 (28.3)	10,333 (26.2)	8803 (22.6)	7061 (17.8)	<0.001
Peripheral artery disease	10,375 (6.8)	4030 (10.6)	2652 (7.0)	2024 (5.4)	1669 (4.4)	<0.001
Chronic obstructive pulmonary disease	4052 (2.7)	1389 (3.7)	1068 (2.8)	924 (2.5)	671 (1.8)	<0.001
Cardiogenic shock	5183 (3.3)	2551 (6.5)	1219 (3.1)	767 (2.0)	646 (1.6)	<0.001
Acute heart failure	6289 (4.0)	2982 (7.7)	1401 (3.5)	948 (2.4)	958 (2.4)	<0.001
Chronic kidney disease	22,402 (14.8)	16,183 (42.7)	4767 (12.5)	1096 (2.9)	356 (0.9)	<0.001
Cardiopulmonary arrest	2948 (1.9)	1379 (3.5)	729 (1.8)	463 (1.2)	377 (0.9)	<0.001
Presentation or diagnosis						
Acute coronary syndrome	60,594 (38.4)	14,799 (38.0)	13,423 (33.8)	14,474 (37.0)	17,898 (44.8)	<0.001
ST-elevation myocardial infarction	28,294 (17.9)	6636 (17.0)	6069 (15.3)	6666 (17.0)	8923 (22.3)	<0.001
Non-ST-elevation myocardial infarction	8350 (5.3)	2310 (5.9)	1797 (4.5)	1864 (4.8)	2379 (6.0)	<0.001
Laboratory values						
Baseline hemoglobin (g/dL)	13.3 ± 1.9	11.8 ± 1.9	13.2 ± 1.6	13.9 ± 1.6	14.3 ± 1.6	<0.001
Baseline creatinine (mg/dL)	0.9 ± 0.3	1.3 ± 0.4	0.9 ± 0.1	0.8 ± 0.1	0.7 ± 0.1	<0.001
Hemoglobin to creatinine ratio	15.9 ± 5.0	9.6 ± 2.2	14.2 ± 0.9	17.3 ± 0.9	22.2 ± 2.6	<0.001
Access site						
Transfemoral intervention	35,019 (22.2)	10,400 (26.6)	8690 (21.8)	7960 (20.3)	7969 (19.9)	<0.001
Transradial intervention	115,670 (73.2)	26,296 (67.4)	29,362 (73.8)	29,600 (75.6)	30,412 (76.0)	
Others	7289 (4.6)	2340 (6.0)	1730 (4.3)	1602 (4.1)	1617 (4.0)	
Number of diseased vessels						
1-vessel disease	97,835 (61.9)	22,018 (56.4)	24,463 (61.5)	25,080 (64.0)	26,274 (65.7)	<0.001
2-vessel disease	40,569 (25.7)	10,803 (27.7)	10,375 (26.1)	9731 (24.8)	9660 (24.2)	<0.001
3-vessel disease	19,100 (12.1)	5994 (15.4)	4806 (12.1)	4280 (10.9)	4020 (10.1)	<0.001
Left main trunk lesion	6504 (4.1)	2262 (5.8)	1700 (4.3)	1408 (3.6)	1134 (2.8)	<0.001
Target coronary artery						
Right coronary artery	52,709 (33.4)	14,126 (36.2)	13,441 (33.8)	12,856 (32.8)	12,286 (30.7)	<0.001
Left main trunk-Left anterior descending artery	83,220 (52.7)	20,009 (51.3)	20,602 (51.8)	20,647 (52.7)	21,962 (54.9)	<0.001
Left circumflex artery	39,471 (25.0)	10,095 (25.9)	10,199 (25.6)	9636 (24.6)	9541 (23.9)	<0.001
Bypass graft	672 (0.4)	263 (0.7)	190 (0.5)	126 (0.3)	93 (0.2)	<0.001
Devices						
Drug-eluting stent	135,601 (85.8)	32,948 (84.4)	34,070 (85.6)	33,787 (86.3)	34,796 (87.0)	<0.001
Bare-metal stent	1709 (1.1)	518 (1.3)	419 (1.1)	387 (1.0)	385 (1.0)	<0.001
Drug-coated balloon	20,195 (12.8)	5282 (13.5)	5366 (13.5)	4960 (12.7)	4587 (11.5)	<0.001
Rotational atherectomy	4887 (3.1)	1588 (4.1)	1273 (3.2)	1093 (2.8)	933 (2.3)	<0.001
Preprocedural medications						
Aspirin	140,382 (88.9)	34,560 (88.5)	35,686 (89.7)	34,926 (89.2)	35,210 (88.0)	<0.001
Clopidogrel	53,941 (34.1)	14,584 (37.4)	14,073 (35.4)	13,065 (33.4)	12,219 (30.5)	<0.001
Prasugrel	79,549 (50.4)	18,136 (46.5)	20,021 (50.3)	20,171 (51.5)	21,221 (53.1)	<0.001
Ticagrelor	123 (0.1)	33 (0.1)	37 (0.1)	29 (0.1)	24 (0.1)	0.375
Oral anticoagulants	11,263 (7.1)	4292 (11.0)	3104 (7.8)	2258 (5.8)	1609 (4.0)	<0.001

Values are presented as *n* (%) or mean ± standard deviation, as indicated.

**Table 2 jcm-09-03612-t002:** In-hospital outcomes.

Outcomes		Hemoglobin to Creatinine Ratio (Hemoglobin (g/dL)/Creatinine (mg/dL))				
	Total Cohort	Quartile 1 <12.5	Quartile 2 12.5 ≤ … < 15.8	Quartile 3 15.8 ≤ … < 19.0	Quartile 4 19.0≤	*p*-Value
	*n* = 157,978	*n* = 39,036	*n* = 39,782	*n* = 39,162	*n* = 39,998	
Procedural success (final TIMI III flow)	155,017 (98.1)	38,171 (97.8)	39,051 (98.2)	38,485 (98.3)	39,310 (98.3)	<0.001
In-hospital mortality	1296 (0.82)	786 (2.01)	243 (0.61)	141 (0.36)	126 (0.32)	<0.001
Cardiac tamponade	230 (0.15)	76 (0.19)	53 (0.13)	48 (0.12)	53 (0.13)	0.032
Cardiogenic shock	1636 (1.04)	729 (1.87)	385 (0.97)	277 (0.71)	245 (0.61)	<0.001
Emergency operation	141 (0.09)	50 (0.13)	33 (0.08)	32 (0.08)	26 (0.07)	0.022
Bleeding complications	491 (0.31)	258 (0.66)	90 (0.23)	79 (0.20)	64 (0.16)	<0.001
Access-site bleeding	246 (0.16)	115 (0.29)	48 (0.12)	44 (0.11)	39 (0.10)	<0.001
Non-access-site bleeding	256 (0.16)	147 (0.38)	44 (0.11)	37 (0.09)	28 (0.07)	<0.001

Values are presented as *n* (%). TIMI = Thrombolysis in Myocardial Infarction.

**Table 3 jcm-09-03612-t003:** Multivariable logistic regression analyses.

Variables	In-Hospital Mortality			Bleeding Complications		
	Odds Ratio	95% Confidence Interval	*p*-Value	Odds Ratio	95% Confidence Interval	*p*-Value
Age	1.02	1.02–1.03	<0.001	1.01	1.00–1.02	0.040
Hgb/Cr ratio	0.91	0.89–0.92	<0.001	0.92	0.90–0.94	<0.001
Oral anticoagulants	1.09	0.83–1.42	0.539	1.66	1.24–2.22	0.006
Female sex	1.69	1.45–1.97	<0.001	2.64	2.15–3.23	<0.001
History of heart failure	1.34	1.12–1.61	0.002	1.19	0.94–1.52	0.148
Acute heart failure	2.80	2.33–3.37	<0.001	1.09	0.78–1.51	0.620
Presentation						
ST-elevation myocardial infarction	4.26	3.57–5.08	<0.001	1.49	1.16–1.90	0.001
Non-ST-elevation myocardial infarction	2.94	2.31–3.74	<0.001	1.62	1.15–2.27	0.005
Cardiogenic shock	7.03	5.82–8.48	<0.001	2.98	2.15–4.14	<0.001
Hypertension	0.68	0.59–0.79	<0.001	0.94	0.75–1.18	0.596
Hyperlipidemia	0.55	0.47–0.63	<0.001	0.74	0.61–0.90	0.003
Diabetes mellitus	1.06	0.92–1.22	0.389	0.79	0.65–0.96	0.017
3-vessel disease	1.59	1.35–1.87	<0.001	1.13	0.88–1.45	0.339
Left main trunk lesion	1.73	1.40–2.12	<0.001	1.45	1.05–1.99	0.024
Access site						
Transfemoral intervention (reference)	-	-	-	-	-	-
Transradial intervention	0.41	0.35–0.49	<0.001	0.30	0.24–0.37	<0.001
Other	0.97	0.72–1.30	0.825	0.75	0.50–1.12	0.155

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
