# Peer review of "Association of the Hemoglobin to Serum Creatinine Ratio with In-Hospital Adverse Outcomes after Percutaneous Coronary Intervention among Non-Dialysis Patients: Insights from a Japanese Nationwide Registry (J-PCI Registry)"

_jcm, 2020, doi:10.3390/jcm9113612_

Round 1

Reviewer 1 Report

The present study by Numasawa et al. investigated the association of the hemoglobin to serum creatinine (Hgb/Cr) ratio with adverse clinical outcomes in 157.978 non-dialysis patients included in the nationwide J-PCI registry. The authors demonstrated a significant and independent relation between Hgb/Cr ratio with both in-hospital mortality and bleeding complications. Overall, the manuscript is interesting and well-written, and the statistical analyses are appropriate. However, the following points should be addressed: 

  • Methodology: retrospective analysis of registry data. Based on the study design, no conclusions can be drawn about a causal relationship between Hgb/Cr ratio and outcome. Given this fact, the title should be changed from “Impact of … on…”, which somehow feigns causality, to “Association of/between … with/and…”.
  • The lack of serial biomarker measurements (particularly creatinine, acute kidney injury as dynamic information) in the acute setting, especially in the subgroup with ACS, is a limitation.
  • To better understand the findings, further subgroup analyses would be interesting. For example, primary findings in patients with ACS vs. stable CAD or in the patient group without cardiogenic shock?   
  • Limited novelty: Both hemoglobin and creatinine have already been demonstrated to provide important prognostic information in these patients (e.g. earlier publication of this very research group: Numasawa et al. Am J Cardiol 2018 Mar 15;121(6):695-702).

Reviewer 2 Report

Numasawa and co-authors described their original manuscript “Impact of the hemoglobin to serum creatinine ratio on in-hospital adverse outcomes after percutaneous coronary intervention among non-dialysis patients: Insights from a Japanese nationwide registry”.

It is important to assess risk factors for PCI and both hemoglobin and renal function are well known as important predictors of adverse clinical outcomes after PCI. This study attempts to determine whether the value of the hemoglobin to creatinine ratio is related to clinical events after PCI.

Overall, this manuscript is well written and adds incremental useful knowledge to our understanding but I have a few questions and comments.

Major comments

1, As authors mentioned, several studies reported both preprocedural hemoglobin value and renal function are strong predictors for adverse outcomes after PCI. That’s why the result of this study makes sense but the reviewer would like to know whether the Hgb/Cr ratio would be better predictor than the hemoglobin value and the creatinine level themselves. If possible, can you compare those statistically?

2, The percentage of patients with taking oral anticoagulants in Quatile 1 showed higher than others and it meant Quartile 1 included lots of patients with AF. Those patients could take antiarrhythmic drugs. Also, some patients should take proton pump inhibitors (PPI) in preventing for gastrointestinal bleeding. These drugs may affect the results of this study. Can you show the data about the percentages of antiarrhythmic drugs and PPI in Table 1?

3, In Table 3, Hgb/Cr ratio showed inversely relationship with in-hospital mortality and bleeding complications but the result was not strongly because many other baseline clinical characteristics demonstrated a strong relationship. For example, female gender had an odds ratio of 2.69 (1.45 to 1.97) to in-hospital mortality and 2.64 (2.15 to 3.23) to bleeding complications. The authors should not mention the relationship was strong.

Minor comments

1, Table 1

p-value in Acute coronary syndrome; <0/001 → <0.001

2, Table 3

Hb/Cr ratio → Hgb/Cr ratio

Round 2

Reviewer 2 Report

I think the revised manuscript adequately revised and is now acceptable.